# Challenges and Advances in Genome Editing Technologies in *Streptomyces*

**DOI:** 10.3390/biom10050734

**Published:** 2020-05-08

**Authors:** Yawei Zhao, Guoquan Li, Yunliang Chen, Yinhua Lu

**Affiliations:** 1College of Life Sciences, Shanghai Normal University, Shanghai 200234, China; zhaoyawei@sibs.ac.cn; 2College of Bioengineering, Henan University of Technology, Zhengzhou 450001, China; 3School of Food and Biological Engineering, Jiangsu University, Zhenjiang 212013, China; 13775556788@139.com; 4School of Agricultural Equipment Engineering, Zhenjiang, Jiangsu University Zhenjiang 212013, China

**Keywords:** genome editing, CRISPR/Cas, *Streptomyces*, microbial natural products

## Abstract

The genome of *Streptomyces* encodes a high number of natural product (NP) biosynthetic gene clusters (BGCs). Most of these BGCs are not expressed or are poorly expressed (commonly called silent BGCs) under traditional laboratory experimental conditions. These NP BGCs represent an unexplored rich reservoir of natural compounds, which can be used to discover novel chemical compounds. To activate silent BGCs for NP discovery, two main strategies, including the induction of BGCs expression in native hosts and heterologous expression of BGCs in surrogate *Streptomyces* hosts, have been adopted, which normally requires genetic manipulation. So far, various genome editing technologies have been developed, which has markedly facilitated the activation of BGCs and NP overproduction in their native hosts, as well as in heterologous *Streptomyces* hosts. In this review, we summarize the challenges and recent advances in genome editing tools for *Streptomyces* genetic manipulation with a focus on editing tools based on clustered regularly interspaced short palindrome repeat (CRISPR)/CRISPR-associated protein (Cas) systems. Additionally, we discuss the future research focus, especially the development of endogenous CRISPR/Cas-based genome editing technologies in *Streptomyces*.

## 1. Introduction

*Streptomyces* are Gram-positive bacteria that can produce high amounts of secondary metabolites, such as antibiotics (e.g., pristinamycin [1] and daptomycin [2]), immunosuppressants (e.g., rapamycin [3] and FK506 [4]), insecticides (e.g., avermectin [5] and milbemycin [6]) and anti-tumor drugs (e.g., daunorubicin [7] and bleomycin [8]), which are widely used in agriculture and veterinary/human medicine. In the last few decades, the traditional strategy for the discovery of novel natural products (NPs) with medicinal properties has not been effective, which is mainly due to the repeated rediscovery of known chemical compounds [9]. The bioinformatic analysis of the sequenced *Streptomyces* genomes has revealed the presence of a large number of NP biosynthetic gene clusters (BGCs). In each genome, there are approximately 20–50 NP BGCs, which are much more than known NPs isolated previously [10]. These NP BGCs are an unexplored rich reservoir of natural compounds, which can be used for the discovery of novel chemical compounds. However, the majority of these BGCs are not expressed or are poorly expressed (commonly called silent BGCs) in streptomycetes under traditional laboratory culture conditions [11]. Recently, various strategies have been developed to activate these silent BGCs and trigger NP overproduction, which has contributed to scientific research on the identification of potential chemical compound resources in streptomycetes [12]. Generally, these strategies can be broadly classified into the following two main categories: (i) the induction of BGCs expression in native *Streptomyces* hosts that is genetically tractable, which usually involves genetic manipulations of the host genome, such as knocking out the competitive pathways [13] and negative regulatory genes [14], replacing the native promoters with strong promoters [15], and overexpression of positive regulatory genes [16]; (ii) cloning and/or refactoring NP BGCs, followed by transfer into surrogate *Streptomyces* hosts for heterologous expression, which is especially useful for the activation of silent BGCs from genetically intractable streptomycetes [17]. The successful heterologous expression of these NP BGCs is always followed by strain improvement to produce high titers of target compounds. To achieve the activation of silent BGCs in either native or heterologous *Streptomyces* hosts, it is critical to develop highly efficient genome editing technologies.

Traditional genetic manipulation strategies for streptomycetes, including DNA deletion, disruption, and replacement, employ suicide plasmids or plasmids with temperature-sensitive replication origin (e.g., pKC1139), which require the selection and screening of single- and double-crossover recombination events, respectively [18]. The traditional strategy, which is a time-consuming process, has low efficiency for genetic engineering. Additionally, the double-crossover mutants are rarely obtained in streptomycetes, which exhibit weak DNA homologous recombination. To address these limitations, various genome editing technologies have been developed (Figure 1), especially the clustered regularly interspaced short palindromic repeat (CRISPR)/CRISPR-associated protein (Cas)-based tools, which have markedly improved the genetic manipulation of streptomycetes and accelerated NP discovery, strain improvement, and functional genome research [19]. In this review, we summarize the challenges and recent advances in key genome engineering technologies for *Streptomyces* with a special focus on the development of novel technologies based on CRISPR/Cas systems. We also discuss the future research focus, especially the development of genome editing tools based on endogenous CRISPR/Cas systems widely distributed in *Streptomyces*.

## 2. PCR-Targeting System

The PCR-targeting system was first developed for gene knockout in *Escherichia coli*, which is based on the high-efficiency recombination between the target region of the *E. coli* genome and a PCR-amplified selectable marker flanked at both ends by 40–50 bp homologous extensions [20]. The recombination events are mediated by the λ recombination system (λ Red), which includes the Red α, β, and γ proteins from the λ phage [21]. Gust et al. adapted this system for nonpolar and in-frame deletion of genes or gene clusters in *S. coelicolor* [22]. The PCR-targeting system-assisted genetic engineering of *Streptomyces* involves four steps: (i) a cosmid library of the *Streptomyces* strain needs to be constructed; (ii) the target gene within the cosmid is knocked out by the PCR-targeting system in *E. coli*. To facilitate the removal of the selectable marker (e.g., the antibiotic-resistant gene), the antibiotic-resistance cassette is designed to be flanked by the *FRT* or *loxP* sites; (iii) the mutant cosmid with deletion of the target gene is introduced into *Streptomyces* to screen for mutant strains with double-crossover recombination events; (iv) the antibiotic-resistance cassette is removed by inducing the expression of the tyrosine recombinase FLP (FLP-*FRT*) or Cre (Cre-*loxP*). The PCR-targeting system is very efficient for the deletion of genes or gene clusters in several *Streptomyces* strains. However, the PCR-targeting system has the following three limitations, which have limited its wide application: (i) a scar (*FRT* or *loxP* site) is left on the genome of the mutant strain; (ii) a cosmid library of *Streptomyces* must be constructed in advance; (iii) the procedure involves four steps, which are cumbersome and time-consuming.

## 3. Cre-*loxP* Recombination System

The Cre-*loxP* recombination system can be used in combination with the PCR-targeting system for genetic engineering in *Streptomyces* as above. Additionally, it can be independently used to knock out large fragments of DNA in *Streptomyces* [23]. This system involves the following two main steps: (i) two *loxP* sites in the same orientation are introduced into the genome to flank the DNA fragment to be deleted by two single-crossover events; (ii) the expression of the Cre recombinase is induced to delete the DNA fragment between the two *loxP* sites. This method is very effective for knocking out large fragments of DNA and can be used to knock out a DNA fragment with a size larger than 1.4 Mb in the industrial strain, *Streptomyces avermitilis* [23]. Similar to the PCR-targeting system, this system also generates a scar (a *loxP* sequence) on the genome and the genetic engineering procedure is time-consuming.

## 4. I-SceI Meganuclease-Promoted Recombination System

Conventional gene knockout methods often use the temperature-sensitive pKC1139 plasmid or segregationally unstable pJTU1278 plasmid to screen for single- and double-crossover events [24,25]. The single-crossover mutants can be easily screened based on antibiotic resistance. However, double-crossover events are often difficult to be obtained, especially in strains exhibiting weak homologous recombination. I-SceI meganuclease can recognize an 18-bp unique sequence and cause DNA double-strand breaks (DSBs), which promote double-crossover recombination events [26,27]. Lu et al. synthesized the codon-optimized *I-SceI* gene and developed an I-SceI-assisted genome editing technology in *S. coelicolor* [28]. Two BGCs that biosynthesize actinorhodin and undecylprodigiosin were successfully deleted by this technology [28,29]. Compared to the traditional gene deletion method, I-SceI cleavage markedly improves the efficiency of double-crossover events.

## 5. CRISPR/Cas-Based Genome Editing

The CRISPR/Cas system, which serves as an adaptive immune system against invading mobile genetic elements (e.g., phages and plasmids), is widely distributed in archaea and bacteria [30]. CRISPR/Cas system-based genome editing technologies have revolutionized genome engineering [31]. These technologies have been widely employed for genome editing in all kingdoms, including animals [32], plants [33], and microbes [34]. Unlike Cre and I-SceI nuclease-based genome editing, CRISPR/Cas-based technology does not require the pre-integration of a unique enzyme recognition sequence into the genome, but employs a transcribed synthetic guide RNA (sgRNA, the chimera of crRNA and tracrRNA), or only crRNA, to direct Cas proteins to any site on the genome [35]. Of the diverse known Cas endonucleases, the *Streptococcus pyogenes*-derived Cas9, which belongs to Class 2 type II, is the most widely used in *Streptomyces* [36]. Recently, Cpf1 (also known as Cas12a) derived from *Francisella novicida*, which belongs to Class 2 type V, has also been used for engineering streptomycetes [37,38]. Compared to other genome editing technologies, the CRISPR/Cas-assisted technology has several advantages, such as higher efficiency, ease of operation, and lower operation time. The CRISPR/Cas-based genome editing technology has markedly promoted the genetic engineering of streptomycetes [39]. Additionally, several CRISPR/Cas-derived technologies, such as the CRISPR interference (CRISPRi)-mediated gene repression tool based on dCas9 (a nuclease-deficient Cas9 with two mutations of D10A and H840A) [40] or ddCpf1 (a nuclease-deficient Cpf1 with the mutation of E1006A) [37] and the base editors (BEs) for targeted base mutagenesis based on dCas9 or Cas9n (a nickase version of Cas9, with the mutation of D10A) have recently been developed [41,42]. They are powerful supplements to the CRISPR/Cas-based genome editing toolbox in *Streptomyces*. A brief description of the CRISPR/Cas-based technologies for genome editing and gene repression in *Streptomyces* is presented in Figure 2 and Table 1.

### 5.1. Cas9-Based Genome Editing

In 2015, four research groups successively developed CRISPR/Cas9-based genome editing tools, which enabled efficient and rapid genetic manipulation, including deletions of single genes or gene clusters, simultaneous deletions of two genes or gene clusters, and point mutations in *Streptomyces* [43,44,45,46]. Cobb et al. generated two sets of CRISPR/Cas9 genome editing systems, namely pCRISPomyces-1 and pCRISPomyces-2, and achieved the precise deletion of various DNA sizes (ranging from 20 bp–31.4 kb) (including individual genes, double genes simultaneously, and single antibiotic BGCs) with efficiency of 21–100% by homology-directed repair (HDR) in three different *Streptomyces* species [43]. Later, they applied this CRISPR/Cas9 technology to perform promoter knock-in for the activation of silent BGCs of different classes in five native *Streptomyces* hosts and achieved the identification of unique metabolites, including a novel pentangular type II polyketide in *Streptomyces viridochromogenes* [15]. We developed a similar CRISPR/Cas9-based editing system, namely, pKCcas9dO, in *S. coelicolor*. This system was applied for genetic engineering at different levels with high efficiency ranging from 60–100%, including deletion of individual genes (*actII-orf4*, *redD*, and *glnR*) and single antibiotic BGCs (with a size of 21.3, 31.6 and 82.8 kb, respectively) [44]. Furthermore, we achieved simultaneous deletion of two genes (*actII-orf4* and *redD*) and two BGCs (21.3 and 31.6 kb) with an efficiency of 54% and 45%, respectively. Additionally, this system was applied to introduce point mutations into the *rpsL* gene with an efficiency of 64%. Around the same time, Tong et al. established the CRISPR/Cas9 editing system (pCRISPR-Cas9) for precise deletion of individual genes and multiple genes by HDR in *S. coelicolor* [45]. Moreover, they reconstituted the non-homologous end joining (NHEJ) repair pathway by introducing the *ligD* gene from *Streptomyces carneus* and generated the editing plasmid pCRISPR-Cas9-ScaligD. This tool acheived high-efficiency inactivation of target genes in the absence of homologous DNA templates. All these genome editing systems, including pCRISPomyces-1, pCRISPomyces-2, pKCcas9dO, and pCRISPR-Cas9, are developed based on plasmids harboring pSG5, a temperature-sensitive replicon [50]. To obtain plasmid-free mutants and perform multiple rounds of genome editing, the editing plasmids must be cured by at least two or three passages at a high temperature (e.g., 37 °C), which is a time-consuming process, especially for slow-growing streptomycetes. Zeng et al. developed the CRISPR/Cas9 editing system pWHU2653 based on a plasmid with pIJ101, a segregationally unstable replicon. Additionally, the counter-selection marker CodA(sm) (the D314A mutant of cytosine deaminase CodA) was introduced [46], which converts 5-fluorocytosine (5-FC) to the toxic compound 5-fluorouracil (5-FU). The introduction of CodA(sm) simplified plasmid curing. The efficiency of plasmid curing can be improved to 95%, which is higher than that of the pIJ101-derived editing plasmid without CodA(sm) (only 39%).

Recently, two modified CRISPR/Cas9 genome editing tools from pWHU2653 have been developed. Mo et al. constructed the editing system pMWCas9 based on pWHU2653 by replacing the *ermE*p*** promoter of Cas9 with the thiostrepton-inducible promoter *tipA*p, which can significantly enhance the DNA transformation efficiency [47]. Importantly, pMWCas9 was successfully used to delete highly repetitive DNA sequences, such as the *eryAIII* gene from erythromycin polyketide synthase (PKS) in *Saccharopolyspora erythraea*. In contrast, the pCRISPR-Cas9 editing tool failed to delete the target gene, which may be caused by the unpredicted DNA recombination resulting from the pSG5 replicon [51]. Wang et al. developed an updated dual-functional chromogenic-screening CRISPR/Cas9 tool (pQS-*gusA* and pQS-*idgS*) based on pWHU2653 by replacing the counter-selection marker CodA(sm) with two reporter systems, GusA and IdgS [48]. These two reporter systems improved the efficiency of both genome editing and plasmid curing up to 100% after chromogenic screening and further simplified the plasmid curing process. This editing system was also successfully applied for efficient genetic manipulation of the genetically recalcitrant and slow-growing rare actinomycete strain *Verrucosispora* sp. MS100137. This genome editing system achieved an efficiency of up to 100% for the deletion of both the carotenoid BGC (5.5 kb) and the abyssomicin BGC (61 kb). In addition, a modified editing tool was developed from the pCRISPR-Cas9 plasmid by introducing both the *S. carneus*-derived *ligD* (required for NHEJ) and a homology template (required for HDR) into one plasmid, thereby allowing either DNA repair pathway to occur [13]. By designing one sgRNA harboring a conserved protospacer sequence targeting a BGC gene of interest, this system achieved the deletion of genes in the BGCs responsible for the biosynthesis of streptothricin or streptomycin (two of the most frequently rediscovered antibiotics) in 11 actinomycete strains with an efficiency of up to 100%. CRISPR/Cas9-mediated inactivation of commonly found BGCs allows the mining of a greater proportion of actinimycete strain collections (many of them are streptomycetes) for new NPs. 

A major challenge for CRISPR/Cas9-mediated genome editing is the high toxicity of Cas9 to the host derived from the off-target DNA cleavage and non-target DNA binding in the absence of sgRNAs, which hampers its application in streptomycetes with low DNA transformation efficiency. To address this issue, Wang et al. established a modified CRISPR/Cas9 system based on pWHU2653 (named pWHU2653-TRMA) by modulating Cas9 activity at multiple levels: (i) using the inducible promoter *tipA*p to regulate the expression of Cas9 instead of the strong constitutive promoter (at the transcriptional level); (ii) introducing the theophylline-inducible riboswitch and Mag-based blue light-inducible reconstitution system to regulate Cas9 activity at the translational and protein levels, respectively [49]. Additionally, as DSB repair is an ATP-dependent process, the gene encoding the ATP synthase β-subunit AtpD was introduced for overexpression to increase the editing efficiency. In *S. coelicolor*, under non-induction conditions, triple controls of Cas9 could markedly reduce its toxicity and increase DNA transformation efficiency by over 250-fold when compared to pWHU2653. As the counter-selection marker CodA(sm) is not suitable for *Streptomyces* with high resistance, they also developed a similar system pKC1139-TRMA based on pKC1139. The pKC1139-TRMA was used to achieve individual deletion of *actII-orf4* and *redD* in *S. coelicolor* with efficiencies ranging from 35% to 80% after simultaneous induction with thiostrepton, theophylline, and blue light for Cas9 activity reconstitution. The pWHU2653-TRMA and pKC1139-TRMA editing systems can achieve the uncoupling of DNA transformation and Cas9-mediated DNA cleavage, which will markedly improve the genetic engineering of *Streptomyces* species with low DNA transformation efficiency.

The genomes of *Streptomyces* strains contain many multicopy genes and mobile genetic elements. The editing of these genes is very difficult because they have identical or highly similar DNA sequences. To address this issue, Najah et al. developed a generic two-step CRISPR/Cas9 editing tool [52], which is similar to I-SceI meganuclease-assisted genome editing technology [28]. First, a non-replicative plasmid (bait DNA) containing the homologous arms flanking the target gene was integrated into the genome by single-crossover recombination. Next, another plasmid was introduced to express Cas9 and sgRNA to cleave the bait DNA (such as the antibiotic-resistance gene), which induced the double-crossover recombination events. The native copies of two xenogeneic silencers *lsr2* paralogs were deleted in *Streptomyces ambofaciens* using this technology. This approach can be widely used for specifically editing one copy of multicopy genes, as well as for the exploration of gene essentiality in *Streptomyces*.

### 5.2. Cpf1-Assisted Genome Editing

The CRISPR/Cas9-based genome editing tools presented above have some limitations that must be addressed. For instance, a G-rich protospacer-adjacent motif (PAM) sequence (5′-NGG-3′) is required for target sequence recognition by Cas9. Due to the high GC content of the *Streptomyces* genome (>70%), the PAM sequence is frequently distributed across *Streptomyces* genomes (e.g., 260 targets per 1000 bp in *S. coelicolor*) [37]. However, the PAM sequence may not be present in the AT-rich DNA regions. Furthermore, to implement multiplex gene editing, the process of plasmid construction for independent transcription of multiple sgRNAs (each has its promoter and terminator) and the introduction of homologous DNA templates for DSB repair is complex and time-consuming. To address these limitations, Cpf1 from *Francisella novicida* (FnCpf1) that recognizes T-rich PAM sequences (5′-TTV-3) has been developed for *Streptomyces* genome engineering by our group [37]. Cpf1 has RNase activity for pre-crRNA processing and the expression of multiple guide crRNAs requires only one promoter, which is an advantage for multiplex genome editing [53]. We achieved the precise deletion of single genes or double genes simultaneously based on HDR with high-efficiency (75–95%) in *S. coelicolor*. Furthermore, we introduced the codon-optimized *Mycobacterium smegmatis ligD* and *ku* genes to reconstitute the NHEJ pathway. The inactivation of target genes or gene clusters by NHEJ-assisted DSB repair can result in random-sized DNA deletions. Further, we observed that Cas9 from *S. pyogenes* and Cpf1 from *F. novicida* exhibited different suitability in the seven tested *Streptomyces* species. Using FnCpf1, we achieved gene deletion in the 5-oxomilbemycin A3/A4-producing strain *Streptomyces hygroscopicus* SIPI-KF, which cannot be edited by Cas9 due to its high toxicity. Similarly, as many *Streptomyces* strains cannot be edited by *S. pyogenes* Cas9, Yeo et al. tested several alternative CRISPR-Cas systems based on the pCRISPomyces-2 system [38]. They demonstrated that Cas9 from *Streptococcus thermophilus* CRISPR1 (Sth1Cas9, PAM: NNAGAA and NNGGAA), Cas9 from *Staphylococcus aureus* (SaCas9, PAM: NNGRRT), and Cpf1 from *F. novicida* (FnCpf1) are functional in multiple streptomycetes, which enables efficient HDR-mediated DNA knock-in and gene deletion. The Cpf1- and alternative Cas9-assisted genome editing technologies can efficiently edit strains that cannot be edited by Cas9 from *S. pyogenes*, such as *Streptomyces* sp. NRRL S-244. Therefore, they are a good complement to the current Cas9-based tools. Hence, a diverse CRISPR/Cas toolbox will markedly facilitate NP discovery and overproduction in *Streptomyces* as well as other actinomycetes.

### 5.3. dCas-Based Transcriptional Repression (CRISPRi)

The CRIPSRi tool based on the nuclease-defective Cas nuclease (such as dCas9 and ddCpf1) can be employed for efficient gene repression by hindering transcription initiation and elongation, which has been demonstrated to be a powerful tool for functional genome research and metabolic engineering in bacteria [54]. Recently, three different CRISPRi tools have been developed in *Streptomyces*. Tong et al. developed an inducible CRISPRi based on genome editing system pCRISPR-Cas9 by replacing Cas9 with dCas9, which enabled efficient repression of single genes upon induction [45]. In this CRISPRi system, the replicative plasmid pGM1190, which harbors a temperature-sensitive replicon pSG5, was used to express the dCas9/sgRNA complex and the dCas9 gene is regulated by the thiostrepton-inducible promoter (*tipA*p). Our research group developed two CRISPRi tools in *S. coelicolor* based on dCas9 [40] and ddCpf1 [37]. In these two systems, pSET152, an integrative plasmid, was used for the expression of the dCas9/sgRNAs or ddCpf1/crRNAs complex and both dCas9 and sgRNAs or ddCpf1 and crRNAs were designed to be regulated by constitutive promoters. Using these two CRISPRi systems, we achieved simultaneous repression of up to four genes at high efficiency. Compared to the inducible CRISPRi tool based on the replicative plasmid (e.g., pGM1190), the latter two systems may have two advantages. First, their repression effects are likely to be stable as they are integrated into the genome. Second, they have wider application, because the efficiency of pSET152 transformation is higher than that of the replicative plasmids. Meanwhile the ϕC31 *attB* site for pSET152 integration is present on the genome of all *Streptomyces* strains (whose genome sequences are available so far) [55]. It is important to note that simultaneous repression of multiple targets using the dCas9-based system involves a time-consuming procedure to construct multiple sgRNA expression cassettes with independent promoters and terminators. In contrast, only a single customized CRISPR array with one promoter is required owing to the pre-crRNA processing ability of ddCpf1, which is time-saving and convenient. Therefore, the ddCpf1-based CRISPRi system has an advantage over dCas9-based systems for multiplexed gene repression.

### 5.4. Base Editors Based on the Cas9 Variants (dCas9 or Cas9n)

CRISPR-guided BEs are emerging genome editing technologies developed in recent years, which enable efficient targeted single-nucleotide-resolution DNA mutagenesis in the genome and are novel powerful tools for genome editing in mammalian cells [56], animals [57], plants [58], and bacteria [59]. The basic principle of BEs is to fuse a Cas9 variant, such as dCas9 (D10A and H840A) or Cas9n (D10A), with a base deaminase to deaminate the exocyclic amine of the target bases, thereby leading to base substitutions [60]. In contrast to the CRISPR/Cas-based genome editing tools mentioned above, BEs do not create DNA DSBs and do not rely on cellular HDR or NHEJ DNA repair pathways. Therefore, it reduces the by-products related to DSBs, such as small insertions or deletions (indels) [61]. Currently, the following two kinds of DNA BEs have been established: cytosine base editors (CBEs) and adenine base editors (ABEs) [62]. CBE can convert cytidine (C) to thymidine (T), while ABE can be used to convert adenosine (A) to guanosine (G). Both types of BEs have been developed for genome editing in *Streptomyces* [41,42].

Tong et al. developed two base editing systems, CRISPR-cBEST (belonging to CBE) and CRISPR-aBEST (belonging to ABE), by fusing rat APOBEC1 (rAPOBEC1) cytidine deaminase and the adenosine deaminase ecTadA to the N-terminus of the codon-optimized Cas9n [41]. To inhibit the activity of the uracil-DNA glycosylase (UDG) and increase the editing efficiency, a codon-optimized uracil glycosylase inhibitor (UGI) from *Bacillus phage* AR9 is linked to the C-terminus of Cas9n in CRISPR-cBEST. In *S. coelicolor*, CRISPR-cBEST converted cytidine to thymidine within a 7-base target window (−11 to −17 bp upstream of the PAM sequence) with frequencies up to 100%. CRISPR-aBEST can convert adenosine to guanosine within a 6-base target window, −12 to −17 bp upstream of PAM. The editing of CRISPR-cBEST follows the priority of TC > CC > AC > GC, while that of CRISPR-aBEST follows the priority of TA > GA > AA > CA. Compared to CRISPR-aBEST, CRISPR-cBEST had higher editing efficiency and off-target effects. Using CRISPR-cBEST, the authors achieved the precise introduction of STOP codons into the designed DNA locations at high frequencies of 60% to 100% in *Streptomyces griseofuscus* and simultaneous targeted mutagenesis of two identical gene copies of the gene *kirN* in *Streptomyces collinus* Tü365. Finally, by introducing the Csy4-based RNA processing system between sgRNA for the expression of multiple sgRNAs with one promoter and terminator, the authors achieved simultaneous editing of three different sites at frequencies up to 100%. Recently, our group has also developed a CBE system dCas9-CDA-UL*_str_* derived from the CBE dCas9-CDA-UL established in *E. coli* [63], which comprises dCas9, PmCDA1 (an activation-induced cytidine deaminase (AID) ortholog from Petromyzon marinus), UGI, and the degradation tag (LVA) [42]. Using dCas9-CDA-UL*_str_*, we achieved single-, double-, and triple-point mutations (cytidine to thymidine) at target sites in *S. coelicolor* with high efficiencies of up to 100%, 60%, and 20%, respectively. This CBE was also applicable for highly efficient base editing in the industrial strain, *Streptomyces rapamycinicus*. Compared to CRISPR-cBEST, which has a 7-base editing window, dCas9-CDA-UL*_str_* has a 5-base editing window, −16 to −20 bp upstream of the PAM sequence. Moreover, dCas9-CDA-UL*_str_* had higher editing efficiency (70–100%) for cytidines preceded by guanosines than rAPOBEC1-derived CRISPR-cBEST (0–60%), which is an advantage for base editing in streptomycetes with high GC contents in their genomes. The development of BEs provides alternative and powerful tools for genome editing in *Streptomces*. In particular, four amino acid codons, namely, Arg (CGA), Gln (CAA and CAG), and Trp codons (TGG, target C in the non-coding strand) can be efficiently mutated to STOP codons (TGA, TAA, and TAG) using CBEs, which results in the inactivation of gene function [64]. Compared to HDR-mediated gene deletion, CBE-based introduction of STOP codons is time- and labor-saving without the cloning of the repair templates. Therefore, we believe that CBEs will significantly facilitate functional genome research and metabolic engineering-based strain improvement in streptomycetes, especially those with weak HR ability.

## 6. Conclusions and Perspectives

The activation of silent BGCs in either native or heterologous *Streptomyces* hosts for NPs discovery normally require extensively genetic manipulation. Therefore, it is critical to develop high-efficiency genome editing tools. Recently, the development and application of the CRISPR/Cas system (particularly, *S. pyogenes* Cas9)-assisted genome editing technologies have markedly improved *Streptomyces* genome engineering, which has facilitated genome mining for novel NP discovery and strain improvement for NP overproduction. However, their application for genome editing requires a high DNA transformation frequency due to the toxic effect of *S. pyogenes* Cas9, as well as the relatively large DNA size of the editing plasmids (normally >10 kb). Therefore, their application is a big challenge in streptomycetes with low DNA transformation ability. To address this issue, researchers have recently adopted two strategies: (i) uncoupling the process between DNA transformation and Cas9-based DNA cleavage by inducing Cas9 expression at the transcription, translation, and protein levels [49]; (ii) using alternative CRSIPR/Cas systems, such as *S. thermophilus* Cas9, *S. aureus* Cas9, and *F. novicida* Cpf1, which have less lethal effects [37,38]. However, owing to the infrequent distribution of PAMs in the *Streptomyces* genome recognized by these alternative Cas nucleases, they may not be as widely used as *S. pyogenes* Cas9. In the future, the following two strategies may be employed to address the toxicity of Cas9.

(i) Developing the Cas9 nickase (Cas9n)-based genome editing technology. Cas9n only causes single-stranded DNA breaks, which can mitigate the lethal effects of Cas9. Currently, Cas9n has been adopted for the development of efficient and precise genome editing tools in several bacteria, such as *Clostridium cellulolyticum* [65], *Pseudomonas putida* [66], *Escherichia coli* [67], *Lactobacillus casei* [68], and *Bacillus licheniformis* [69]. However, Cas9n has not been adopted for *Streptomyces* genome editing. We believe that the Cas9n-assisted genome editing technology may be suitable for engineering streptomycetes with low DNA transformation efficiency.

(ii) Developing genome editing tools based on endogenous CRISPR/Cas systems. CRISPR/Cas systems are reported in approximately 45% of bacterial genomes, including streptomycetes, which are a rich source for the development of endogenous CRISPR/Cas nuclease-based genome editing tools [30]. In addition to mitigating the toxicity of heterologous Cas nucleases by harnessing endogenous CRISPR/Cas systems, the DNA transformation frequency can be improved as the size of the editing plasmid is approximately 4.2 kb (the size of the *S. pyogenes cas9* gene) smaller than that based on heterologous Cas9. Therefore, endogenous CRISPR/Cas machinery provides a superior foundation for genome editing by precluding heterologous *cas9* expression. For example, Pyne et al. reported the detailed characterization of a functional native Type I-B CRISPR/Cas system in *Clostridium pasteurianum* [70]. This CRISPR/Cas system was repurposed for precise markerless genome editing. Compared to the widely employed *S. pyogenes* type II CRISPR-Cas9 system that generated 25% of the total yield of edited cells, the endogenous Cas system-based tool could achieve gene deletion with an efficiency of 100% by only introducing a synthetic crRNA array and templates for DSB repair. In addition to *C. pasteurianum*, endogenous CRISPR-Cas systems have been repurposed for genome editing in various bacteria, including *Escherichia coli* [71], *Pectobacterium atrosepticum* [72], *Streptococcus thermophilus* [73], *Streptococcus mutans* [74], *Lactobacillus crispatus* [75]*,* and *Clostridium tyrobutyricum* [76]. Recently, Zhang et al. comparatively analyzed the wide distribution of native CRISPR/Cas system in the genus *Streptomyces* [77]. The homologous *cas* gene clusters were searched in the genome sequences of 46 *Streptomyces* species harboring the CRISPR loci. The analysis revealed that 26 strains contained 29 *cas* gene clusters. Of these, 26 clusters belonged to the type I-E subtype, two clusters belonged to I-C types, and one cluster belonged to the I-U subtype. Additionally, Qiu et al. identified an active type I-E CRISPR-Cas system with a completely conserved PAM sequence (5′-AAG-3′) in the industrial strain *S. avermitilis* [78]. By repurposing the CRISPR/Cas system, they achieved strain protection from the infection of phages with target protospacers. In the future, endogenous CRISPR/Cas systems in streptomycetes can be readily applied for genome editing by only delivering an engineered synthetic CRISPR array or a small RNA guide in streptomycetes with a limited activity of heterologous CRISPR/Cas systems. However, the specific PAM sequences and the activity of the endogenous Cas nucleases must be fully characterized before repurposing them for genome editing, which can be achieved by plasmid interference assays combined with bioinformatics analysis (e.g., CRISPRTarget) [79].

## Figures and Tables

**Figure 1 biomolecules-10-00734-f001:**
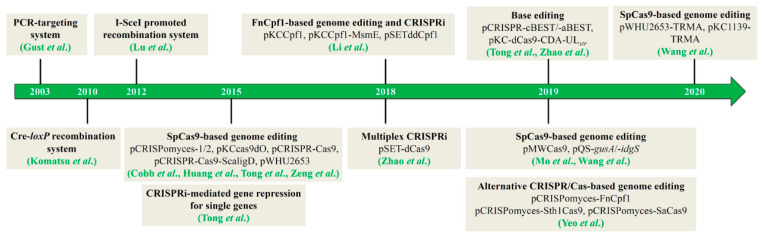
The development course of genome editing technologies in *Streptomyces*.

**Figure 2 biomolecules-10-00734-f002:**
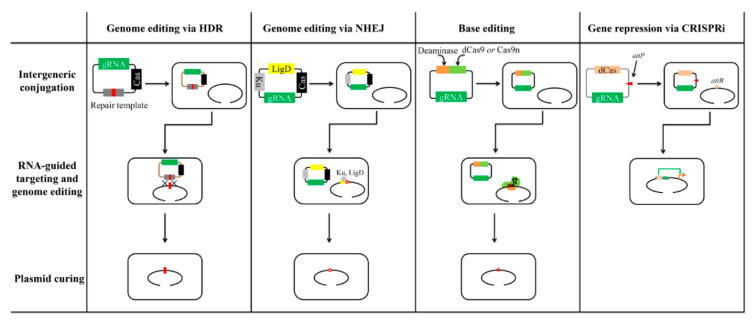
A brief illustration of the CRISPR/Cas system-based technologies for genome editing in *Streptomyces*. Column 1: Genome editing by homology-directed repair (HDR). An editing plasmid with the expression cassettes of the Cas endonuclease and a small guide RNA (gRNA), and containing homologous repair template is introduced into *Streptomyces* by intergeneric conjugation. The Cas nuclease cleaves the target site on the genome with the help of gRNA, resulting in double-strand break (DSB). The DSB is repaired by HDR in the presence of homologous repair template. Then, specific mutations, such as deletions, insertions, and point mutations, are introduced. After plasmid curing, the edited strain could be subject to next round genome editing. Column 2: Genome editing by non-homologous end joining (NHEJ). An editing plasmid containing the expression cassettes of the Cas endonuclease, a small gRNA, and the *ku*/*ligD* genes is introduced into *Streptomyces* by intergeneric conjugation. After RNA-guided DNA cleavage by the Cas nuclease, the DSB is repaired by the action of LigD and Ku. Subsequently, the editing plasmid is cured to facilitate next round genome editing. Column 3: Base editor. The gene encoding a deaminase is fused with the *dCas9* or *Cas9n* gene to induce base substitutions. After introduction of the editing plasmid into *Streptomyces* by intergeneric conjugation, the fusion protein could achieve RNA-guided base editing within a specific target window. After plasmid curing, the edited strain could be subject to next round genome editing. Column 4: Gene repression by CIRPSRi. An integrative plasmid with the expression cassettes of the nuclease-deficient *Cas* (*dCas*) gene and gRNA is introduced into *Streptomyces* by conjugation. After site-specific recombination, the plasmid is integrated into the genome. dCas/gRNA complex could repress the transcription of target genes by interfering with transcriptional initiation or elongation.

**Table 1 biomolecules-10-00734-t001:** The clustered regularly interspaced palindrome repeat (CRISPR)/CRISPR-associated protein (Cas) system–based plasmids developed for genome editing in *Streptomyces.*

Editing Plasmids	Replicons	Cas Proteins	Origins	Promoters of Cas Proteins	Promoters of Guide RNA	Editing Efficiency	Other Features	Addgene Number or Source of Plasmid Request	Reference
pCRISPomyces-1	pSG5	Cas9	*Streptococcus pyogenes*	*rpsL*p(XC)	*rpsL*p(CF)-tracrRNA *gapdh*p(EL)-crRNA	21–25%	-	61736	[43]
pCRISPomyces-2	pSG5	Cas9	*Streptococcus pyogenes*	*rpsL*p(XC)	*gapdh*p(EL)-sgRNA	67–100%	-	61737	[43]
pKCcas9dO	pSG5	Cas9	*Streptococcus pyogenes*	*tipA*p	*j23119*-sgRNA	29–100%	-	62552	[44]
pCRISPR-Cas9	pSG5	Cas9	*Streptococcus pyogenes*	*tipA*p	*ermE*p***-sgRNA	3–100%	-	125686	[45]
pCRISPR-Cas9-ScaligD	pSG5	Cas9	*Streptococcus pyogenes*	*tipA*p	*ermE*p***-sgRNA	69–77%	LigD	125688	[45]
pCRISPR-dCas9	pSG5	dCas9	*Streptococcus pyogenes*	*tipA*p	*ermE*p***-sgRNA	ND	-	125687	[45]
pWHU2653	pIJ101	Cas9	*Streptococcus pyogenes*	*aac(3)IV*p	*ermE*p***-sgRNA	93–99%	CodA(sm)	Yuhui Sun group	[46]
pMWCas9	pIJ101	Cas9	*Streptococcus pyogenes*	*tipA*p	*ermE*p***-sgRNA	ND	CodA(sm)	Xudong Qu group	[47]
pQS*-gusA*	pIJ101	Cas9	*Streptococcus pyogenes*	*tipA*p	*ermE*p***-sgRNA	100%	GusA	Chengzhang Fu group	[48]
pQS-*idgS*	pIJ101	Cas9	*Streptococcus pyogenes*	*tipA*p	*ermE*p***-sgRNA	100%	IdgS	Chengzhang Fu group	[48]
pWHU2653-TRMA	pIJ101	Cas9	*Streptococcus pyogenes*	*tipA*p	*ermE*p***-sgRNA	8.3–80%	AtpD	Xuming Mao group	[49]
pKC1139-TRMA	pSG5	Cas9	*Streptococcus pyogenes*	*tipA*p	*ermE*p***-sgRNA	8.3–80%	AtpD	Xuming Mao group	[49]
pKCCpf1	pSG5	Cpf1	*Francisella novicida*	*ermE*p***	*kasO*p*-crRNA	75–95%	-	Yinhua Lu group	[37]
pKCCpf1-MsmE	pSG5	Cpf1	*Francisella novicida*	*ermE*p***	*kasO*p*-crRNA	10–56.7%	LigD, Ku	Yinhua Lu group	[37]
pSETddCpf1	-	ddCpf1	*Francisella novicida*	*ermE*p***	*kasO*p*-crRNA	11.8–95.2%	-	Yinhua Lu group	[37]
pCRISPomyces-Sth1Cas9	pSG5	Cas9	*Streptococcus thermophilus*	*rpsL*p(XC)	*gapdh*p(EL)-sgRNA	100%	-	129552	[38]
pCRISPomyces-SaCas9	pSG5	Cas9	*Staphylococcus aureus*	*rpsL*p(XC)	*gapdh*p(EL)-sgRNA	87–100%	-	129553	[38]
pCRISPomyces-FnCpf1	pSG5	Cpf1	*Francisella novicida*	*rpsL*p(XC)	*gapdh*p(EL)-crRNA	87–100%	-	129554	[38]
pSET-dCas9	-	dCas9	*Streptococcus pyogenes*	*ermE*p***	*-*	-	-	110183	[40]
pSET-dCas9-actII-4-NT-S1	-	dCas9	*Streptococcus pyogenes*	*ermE*p***	*j23119*-sgRNA	68–99%	-	110185	[40]
pCRISPR-cBEST	pSG5	Cas9n	*Streptococcus pyogenes*	*tipA*p	*ermE*p***-sgRNA	0–100%	rAPOBEC1	125689	[41]
pCRISPR-aBEST	pSG5	Cas9n	*Streptococcus pyogenes*	*tipA*p	*ermE*p***-sgRNA	0–100%	ecTadA	131464	[41]
pKC-dCas9-CDA-UL*_str_*	pSG5	dCas9	*Streptococcus pyogenes*	*tipA*p	*j23119*-sgRNA	15–100%	PmCDA1	Yinhua Lu group	[42]

ND, not determined.

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
