# Peer review of "Challenges and Advances in Genome Editing Technologies in Streptomyces"

_biomolecules, 2020, doi:10.3390/biom10050734_

Round 1

Reviewer 1 Report

This manuscript is a review of current advances in gene editing tools available for species of Streptomyces. In particular, the content focuses most heavily on CRISPR-related editing systems. The manuscript is well organized, easy to read, and highlights the key developments that, at least in my opinion, are needed to bring the reader up to date in the field. The topic is significant and of great interest to a large community of researchers working on Actinomycetes  and in particular, the natural products produced by these bacteria.

General comments:

Apparently there are two figures and a table that accompany this review, but those were not available to me and have not been reviewed here. I think the figures would be very helpful in delineating some of the key differences between CRISPR systems in use for Streptomyces.

The authors refer to a toolbox and to the different plasmid constructs as useful tools for the field. In addition to a listing in the Table, as indicated by the text, it would be extremely helpful to the community to have information about how and where the different tools can be obtained, i.e. are the available by request from individual laboratories, available through plasmid repositories (as is the case for the pCrispomyces), etc.

A recently published article would be good to include in this review, because it incorporates some new plasmid constructions. The article reference is: Culp et al., Nature Biotechnology, Vol 37: 1149-1154, 2019.

Other comments:

1- Lines 70-78. Label the four steps 1,2,3,4 – the current text doesn’t demarcate the steps well.

2- line 85. “independent used” needs to be corrected

3- line 98. ‘are often difficult to occur’ grammar needs correction

4- lines 145-146. It is unclear from the text what is meant by “deletions of double genes.” I think it would help to be more explicit – deletion of two genes or BGCs within a target genome, or something like that.

5- line 198. The ‘genome of Streptomyces’ What species do you mean or do you mean genomes in general?

6- line 264. Define Bes (base editors, I believe).

7- lines 313-315. Describe the reasons why Cas9 is toxic above, where it is first mentioned (lines 229-231.

Reviewer 2 Report

The authors reviewed the current techniques of genome editing in Streptomyces, the most important bacterial species thus far to produce bioactive metabolites for lead compound discovery. the topic is interesting and worth for publication after careful language editing. However, the language is needed to be significantly improved. The logic connections among the sections/paragraphs/sentences are needed. Below here are some of obvious disconnections/grammar issues:

  1. In the abstract, the discover for new natural products and gene editing techniques lack logical connections, please briefly explain why genome editing can facilitate the new discovery of NPs, as well as in the Conclusion and Perspective section.
  2. In the last paragraph of Conclusion and Perspective, there still are contents about the current development of endogenous CRISPER/Cas system etc., which should be included in the 5th part of the manuscript.
  3. Line 33-34, 20-50 NP BGCs and number of unknown NPs doesn’t correspond, please specify
  4. Line 80, ‘has’ should be changed to ‘have’
  5. Line 85, ‘independent’ should be changed to ‘independently’
  6. Line 109, ‘biology’ is vague and unspecific, please rephrase
  7. Line 110, ‘mammals’ is not a kingdom, please revise
  8. Line 115, grammar, please rephrase
  9. Line 143-144, gene deletion and antibiotic BGCs doesn’t correspond, please rephrase
  10. Line 171, should be ‘duel-functional’
  11. Line 264, please rephrase the title of 5.4
  12. Line 371, delete ‘the’ before ‘researchers have recently…’
